# Morphological, Anatomical, and Physiological Characteristics of Heteroblastic *Acacia melanoxylon* Grown under Weak Light

**DOI:** 10.3390/plants13060870

**Published:** 2024-03-18

**Authors:** Xiaogang Bai, Zhaoli Chen, Mengjiao Chen, Bingshan Zeng, Xiangyang Li, Panfeng Tu, Bing Hu

**Affiliations:** 1Key Laboratory of State Forestry and Grassland Administration on Tropical Forestry, Research Institute of Tropical Forestry, Chinese Academy of Forestry, Guangzhou 510520, China; bxg469808@163.com (X.B.); zlchen0525@163.com (Z.C.); cmjmengjiao1997@163.com (M.C.); b.s.zeng@vip.tom.com (B.Z.); xiangylxiang@163.com (X.L.); 2College of Agriculture and Biology, Zhongkai University of Agriculture and Engineering, Guangzhou 510225, China

**Keywords:** heteroblasty, phenotypic plasticity, phyllodes, stress, photosynthesis, *Acacia melanoxylon*

## Abstract

*Acacia melanoxylon* is a fast-growing macrophanerophyte with strong adaptability whose leaf enables heteromorphic development. Light is one of the essential environmental factors that induces the development of the heteroblastic leaf of *A. melanoxylon*, but its mechanism is unclear. In this study, the seedlings of *A. melanoxylon* clones were treated with weak light (shading net with 40% of regular light transmittance) and normal light (control) conditions for 90 d and a follow-up observation. The results show that the seedlings’ growth and biomass accumulation were inhibited under weak light. After 60 days of treatment, phyllodes were raised under the control condition while the remaining compound was raised under weak light. The balance of root, stem, and leaf biomass changed to 15:11:74 under weak light, while it was 40:15:45 under control conditions. After comparing the anatomical structures of the compound leaves and phyllode, they were shown to have their own strategies for staying hydrated, while phyllodes were more able to control water loss and adapt to intense light. The compound leaves exhibited elevated levels of K, Cu, Ca, and Mg, increased antioxidant enzyme activity and proline content, and higher concentrations of chlorophyll a, carotenoids, ABA, CTK, and GA. However, they displayed a relatively limited photosynthetic capacity. Phyllodes exhibited higher levels of Fe, cellulose, lignin, IAA content, and high photosynthetic capacity with a higher maximum net photosynthetic rate, light compensation point, dark respiration rate, and water use efficiency. The comparative analysis of compound leaves and phyllodes provides a basis for understanding the diverse survival strategies that heteroblastic plants employ to adapt to environmental changes.

## 1. Introduction

Unlike animals, plants, particularly woody perennials, cannot move and can only adjust to changes in external conditions through internal modifications. Phenotypic plasticity is therefore fundamental to maintaining optimal fitness when environmental conditions fluctuate or upon exposure to transitory harmful conditions [1]. Heteroblasty in plants refers to a sudden morphological transition during development, as can be observed in *Populus euphratica* [2], *Sabina vulgaris* [3], and *Eucalyptus globulus* [4]. In contrast to heteroblasty, which refers to the transition of leaves associated with age-related development at the leaf level, heterophylly is more likely an adaptation to environmental changes. The phenotypic transition of many aquatic plant species (e.g., *Callitriche heterophylla*, *Rorippa aquatica*, and *Hygrophila difformis*) is often described as heterophylly [5,6,7]. To this day, the words heteroblasty and heterophylly are not well distinguished and are often used interchangeably.

Some research reveals that the formation of heteromorphic leaves may be regulated by ontogenetic development or environmental factors. The leaves of *P. euphratica* in the seedling stage are stripped, but they appear lanceolate, ovate, and broad-ovate with ontogenetic development [2]. The needle and scale leaves of *S. vulgaris* appear at different crown positions and growth stages [3]. Similarly, *E. globulus* completes the transformation from wide and thin juvenile leaves to narrow and thick adult leaves during successive stages of development [4]. Moreover, many aquatic plants are highly susceptible to the complex environment of alternating water and land and have evolved aquatic and terrestrial leaves with widely differing morphologies [5,6,7]. Many genes related to hormone regulation pathways and environmental stress have been screened out in *P. euphratica* [8,9], *Juniperus flaccida*, *Pinus cembroides* [10], and some aquatic plants [11], which are essential in regulating vascular development, stomatal morphology, leaf polarity, and epidermal cell differentiation. Recent studies have employed multi-omics analysis to investigate the regulation and function of heteromorphic leaves [12]. Nevertheless, the exact mechanism of heteromorphic leaves formation has yet to be elucidated.

Leaves are a vital link between plants and their environment [1]. Heteromorphic leaves exhibit distinct anatomical and physiological characteristics which enable them to survive in challenging conditions. The broadly ovate leaves located at the top of the trunk of *P. euphratica* displayed enhanced leaf area, cuticle and cuticular wax thickness, epidermal cell number and length, mucilage cell numbers, and palisade tissue thickness compared to other leaves [13]. They exhibited a higher transpiration rate (E), water use efficiency (WUE), and proline (Pro) and hormone (IAA, GA, ZR) content, as well as a synergistic alteration to sustain a high photosynthetic efficiency [14]. Therefore, the broad-ovate leaves have better adaptability to high temperatures, intense light, and dry environments due to their developed xeric structure, higher photosynthesis, and water retention ability. Similarly, the scale leaves of *S. vulgaris* possessed higher leaf area mass (LMA) and nitrogen content, a higher photosynthetic rate and light compensation point, a higher water use efficiency, stronger resistance to environment stress, a high density of stomata (SD) and a high barrier tissue spongy tissue ratio (P/S), which easily adapts to high radiation temperatures and drought environments compared to needle leaves [3,15]. In aquatic plants, the terrestrial leaves are broader and thicker, with more stomata and well-developed vascular bundles, which enable them to make the most of the sunlight. At the same time, the aquatic leaves are narrower, thinner, and more filamentous, linear, or deeply lobed, enabling them to maximize oxygen and nutrient uptake in the water [16,17]. Despite this, the variations in physiological characteristics and functions between the heteromorphic leaves of different tree species still require further investigation.

*Acacia melanoxylon* B.R. belongs to the genus *Acacia* Mill of the Mimosaceae family. Originating from Australia, the species has a broad natural range, stretching from the Atherton Tableland in the north of Queensland along the eastern coast to Tasmania in the south and into South Australia in the west, spanning latitudes from 16 to 43° S [18,19]. *A. melanoxylon* adapts to various forest types, such as swamps, tropical rainforests, and beaches, with a wide range of adaptability and strong competitiveness [20]. *A. melanoxylon* has typical heteroblastic characteristics of Acacia, with two basic leaf types: pinnate compound leaves and phyllodes. During the early stages of growth, bipinnately compound leaves (known as juvenile leaves or “true leaves”) emerge. Subsequently, the petiole becomes broad and flattened but with compound leaves, forming a transition leaf. After a few weeks to a few years, the compound leaf degenerates and disappears, while the phyllode development matures and becomes the main organ of photosynthesis [18,21,22,23]. The heteroblastic leaf phenotypes were believed to be due to their adaptation to low-light or dry environments [24,25]. Previous studies have been done on the anatomical structure and photosynthetic capacity of *A. melanoxylon* heteroblastic leaves, but there are still many mysteries about the developmental regularity and formation mechanism. In this study, the heteroblastic leaves of *A. melanoxylon* were induced through weak light treatment. Then, we comprehensively evaluated the growth and development difference model of compound leaves and phyllodes by analyzing the differences between the two types of leaves on growth characteristics, anatomical structure, photosynthetic capacity, physiologically active substances content, and antioxidant enzyme activity to research the different survival strategies of plants to cope with different environments and try to explore the intrinsic regulatory mechanism of leaf morphology change.

## 2. Results

### 2.1. Heteroblastic Phenotype and Growth of A. melanoxylon Leaves Induced by Weak Light Treatment

Under the weak light treatment, most new leaves grown within the treatment time were formed as compound leaves, whereas more phyllodes were created in the seedlings under control conditions from treatment for 60 d, while compound leaves were mainly raised before (Figure 1a,d). As shown in Figure 1b,c, the height, ground diameter, and total number of leaves of *A. melanoxylon* seedlings under the weak light treatment were less significant than those under the control treatment. After treatment for 90 days, the seedlings under the weak light treatment had fewer roots, stems, and leaves, as well as less total biomass than those in the control (Table 1). Regarding biomass distribution, the root–shoot ratio of seedlings under the weak light treatment was significantly lower than those under the control. The ratio of compound leaves to total biomass under the weak light treatment was 0.74, while the ratios of phyllodes and compound leaves to total biomass were 0.21 and 0.24 under control conditions, respectively. The balance of root, stem, and leaf biomass was 40:15:45 in the control and 15:11:74 under the weak light treatment. This revealed that weak light treatment maintained the compound leaf phenotype, and the biomass percentage of the compound leaves was about 74%, 4.5 times higher than the roots. The phyllode phenotype arose after treatment for 60 days in the control, and the biomass percentage of phyllodes and compound leaves was about 45%, close to the biomass percentage of roots.

### 2.2. Differences in Anatomical Structure between Heteroblastic Leaves

As shown in Figure 2b,d, the phyllodes epidermis was relatively smooth and covered by a thick cuticle. The phyllodes had dense palisade mesophyll layers under the epidermis on both sides of the parenchymatous tissue in the central region. The phyllodes had multiple vascular bundles, mainly with a large central vascular bundle symmetrically placed above and below and multiple small vascular bundles, all with their xylem facing the central parenchymatous tissue. Both large and small vascular bundles had sturdy fibrous caps. The large vascular bundles extended almost to the epidermis, and several layers of lignified parenchyma cells were situated between the fiber caps and the epidermis (Figure 2b). In contrast, the adaxial surface of the compound leaves had an irregular undulating “hillock” morphology (Figure 2g), with a thin cuticle (Figure 2f) and a plate-type epicuticular wax layer (Figure 2h) adhering to its surface. The compound leaves featured a loosely arranged palisade mesophyll layer beneath the upper epidermis and a dense spongy tissue beneath the lower epidermis. The compound leaves had multiple vascular bundles, mainly a large central bundle and multiple small vascular bundles. Compared to the phyllodes, the vascular bundles of the compound leaves did not have a fibrous cap and were located between the barrier and spongy tissues (Figure 2f).

As shown in Table 2, Figure 2c,g, the stomatal density on the adaxial surface of the phyllodes was significantly higher than on the adaxial surface of the compound leaves at 660 (number of cells per mm^2^). However, the compound leaves’ stomatal width and aperture size were significantly higher than those of the phyllodes.

### 2.3. Differences in the Levels of Mineral Elements and Cell Wall Components between Heteroblastic Leaves

Figure 3 illustrates the accumulation of various elements (N, P, K, Fe, Cu, Zn, Ca, Mg, and B) in the phyllodes and compound leaves after 90 days of treatment. The levels of N, K, Cu, Zn, Ca, Mg, and B contained in the compound leaves were found to be higher than those in the phyllodes. Notably, the contents of K, Cu, Ca, and Mg in the compound leaves were significantly greater than in the phyllodes (Figure 3c,e,g,h) by 63.53%, 83.51%, 302.63%, and 16.22%, respectively (*p* < 0.05). However, the content of Fe was significantly higher in the phyllodes than in the compound leaves, by 105.31% (*p* < 0.05) (Figure 3d). The compound leaves exhibited significantly lower cellulose and lignin contents compared to the phyllodes (*p* < 0.01 or 0.05) (Figure 3k,l), showing decreases of 33.43% and 31.23%. However, the hemicellulose contents did not show significant differences between the two types of leaves (Figure 3j).

### 2.4. Differences in the Levels of Antioxidant Enzyme Activities and Osmotic Regulatory Substances between Heteroblastic Leaves

Results from Figure 4 indicate that Pro content and SOD, POD, and CAT activity in compound leaves were significantly higher than those in phyllodes, with a *p* value of less than 0.01. It is suggested that in weak light conditions, compound leaves accumulate more osmoregulatory substances (Pro) and that this increases the activity of antioxidant enzymes to accommodate the needs of the weak light.

### 2.5. Differences in Light Utilization Capacity between Heteroblastic Leaves

As shown in Figure 5a, the responses of compound leaves and phyllodes Pn to PAR were comparable. When PAR was smaller than 200 μmol·m^−2^·s^−1^, the Pn of compound leaves and phyllodes increased in a straight line. As PAR increased to 200–400 μmol·m^−2^·s^−1^, phyllode Pn increased faster than that of the compound leaves. When PAR was larger than 600 μmol·m^−2^·s^−1^, the Pn of compound leaves and phyllodes increased slowly with PAR enhancement and then gradually stabilized. The light response parameters P′_max_, LCP, and R_d_ of phyllodes were significantly higher than those of compound leaves (*p* < 0.05) (Table 3). At 2000 μmol·m^−2^·s^−1^ PAR, the Ci of compound leaves was significantly higher than those of phyllodes (*p* < 0.05), whereas the WUE was considerably lower than that of phyllodes (*p* < 0.05). It suggests that phyllodes can adapt to stronger light, exhibiting higher photosynthetic and water use capacity. Conversely, compound leaves may encounter challenges in adjusting to low-light environmental conditions because they have a strong intercellular CO_2_ concentration. The comparison of photosynthetic pigment contents revealed that the chlorophyll a and carotenoid contents were significantly higher in compound leaves than in phyllodes (*p* < 0.01).

### 2.6. Differences in the Levels of Endogenous Hormone between Heteroblastic Leaves

Endogenous hormones serve as crucial regulators of plant growth and development by modulating the absorption and utilization of mineral elements, as well as influencing the activity of anti-stress enzymes. The levels of endogenous hormones in the phyllodes and the compound leaves of *A. melanoxylon* were significantly different (*p* < 0.01, Figure 6). The contents of ABA, CTK, and GA in compound leaves were significantly higher than those in the phyllodes, with increases of 9.73%, 35.44%, and 62.79%, respectively (*p* < 0.01). In contrast, the IAA content in the phyllodes was found to be significantly higher than in the compound leaves and increased by 16.92% (*p* < 0.01).

### 2.7. Correlation Analysis of Growth and Development-Related Indicators in Heteroblastic Leaves

The treatment of weak light showed a significant positive correlation with leaf mass ratio, stomatal length, and width, as well as the contents of K, Cu, Ca, Mg, Pro, ABA, CTK, and GA, along with the activities of SOD, POD, and CAT. Significant negative correlations existed with ground diameter, root biomass, stem biomass, leaf biomass, total biomass, root–shoot ratio, phyllodes mass ratio, stomatal width, stomatal density, contents of Fe, cellulose, and lignin, P’_max_, and WUE. From this, we can presume that the low light environment was closely associated with the development of compound leaves and influenced stomatal development, hormone synthesis, mineral element accumulation, and antioxidant enzyme activity. However, weak light was not conducive to the maximum growth of plants. P′_max_ showed a negative correlation with stomatal length and width and levels of Ci, ABA, CTK, GA, K, Cu, Ca, Mg, and Pro, as well as the activities of SOD, POD, and CAT. It showed a positive correlation with stomatal width, stomatal density, and WUE, and the levels of ABA, CTK, and GA were positively associated with the content of K, Cu, Ca, Mg, and Pro, as well as the activities of SOD, POD, and CAT. Mineral elements may interact with hormones and antioxidant enzymes to affect photosynthesis.

## 3. Discussion

### 3.1. Heteroblasty Is Highly Sensitive to the Induction of Weak Light in A. melanoxylon Leaves

The transformation of heteroblastic leaves of *A. melanoxylon* is highly susceptible to the light environment. In the present study, the leaves of the seedlings under the weak light treatment remained compound during the treatment time, whereas seedlings under the control treatment began to grow more phyllodes gradually around 60 days into the treatment (Figure 1a,d). It suggests that decreased irradiance delays the transformation of the heteroblastic leaves. Our results were consistent with those found in studies of *Acacia implexa* [22,26] and *Acacia koa* [27], but there were some differences in the timing of the transition from compound leaves to phyllodes in these species. *A. implexa* undergoes heteroblastic development in about 120 days [22]; *A. koa* usually completes the transition to phyllodes within 3–5 months after seed germination, but this transition may continue for several years when light is low [27,28]. Furthermore, heteroblastic development may represent an optimal strategy in response to resource scarcity. Previous studies have shown that plants can adjust their biomass allocation and morphology in response to resource stress, such as limited light availability [29]. Interestingly, heteroblastic species can modify both overall biomass allocation and localized leaf-level changes, indicating a noteworthy capacity for adaptation. This phenomenon was observed in this experiment, as well as in other Acacia species [26]. The study’s results revealed a significantly lower root–shoot ratio for weak light-treated seedlings than those in the control group (Table 1), indicating a preference for *A. melanoxylon* to allocate more resources to stem and leaf growth in weak light environments. The analysis of the mass ratio between compound leaves or phyllodes to total plant revealed that when the plant was in a weak light climate, more biomass was allocated to the compound leaves (0.74 of whole plant, Table 1), yet the amount of biomass assigned to the phyllodes was increased in the control (0.21 of phyllodes and 0.24 of compound leaves, Table 1). This reveals the adaption strategies of *A. melanoxylon* adaptation to a low-light environment, which involve the formation of a large proportion of compound leaves to enhance energy synthesis. Despite this, weak light continues to restrict seedling growth. For instance, *A. koa* seedlings grown under low-light conditions exhibited relatively small total biomass despite their increased height [27]. The results of the present study show that weak light-treated seedlings exhibited low biomass levels in the roots, stems, leaves, and overall plant, accompanied by poor growth characteristics, including plant height and ground diameter (Figure 1b,c). Finally, *A. melanoxylon* exhibits high plasticity overall, despite its restricted growth in weak light. The retention of compound leaves and greater allocation of biomass to the above-ground portion may represent an optimal adaptive evolutionary strategy for weak light environments. The assessment of this strategy’s inception, development, and ultimate goal is an intriguing prospect.

### 3.2. Anatomical Structural Adaptive Characteristics of A. melanoxylon Heteroblastic Leaves

In contrast to compound leaves, phyllodes exhibit isobilateral leaf anatomy, characterized by a symmetrical arrangement of closely packed palisade tissues above and below the mesophyll, as well as a thick cuticle (Figure 2b). This aligns with the findings of *A. koa* [30]. In addition to the previously known characteristics, compound leaves have been found to feature an epicuticular wax with a plate-like structure on their surface (Figure 2h). Previously, it was established that the high palisade packing of leaves protects against rapid wilting during water stress [31]. The thick cuticle and epicuticular waxy layers are commonly regarded as effective barriers against water loss and as protective shields for leaves against excessive radiation damage [32,33,34]. It is clear that both phyllodes and compound leaves each have their own protective structures. It was observed in *P. euphratica* that broad-ovate leaves positioned in the upper canopy and exposed to greater solar radiation exhibit smaller stomatal sizes and higher stomatal densities compared to lanceolate leaves in the lower canopy [35]. The present study yielded comparable findings, showing reduced stomatal size and increased stomatal density in the phyllodes. These characteristics are likely to enhance the efficient control of transpirational water loss, particularly in conditions of elevated light intensity and temperature (see Figure 2c,d).

The degree of vascular bundle development also indicates heteroblastic leaves’ different adaptive capacities. Within the heteroblastic leaves of *P. euphratica*, the broad-ovate leaves exhibit more pronounced vascular bundles and demonstrate superior drought resistance to other leaf types [14]. Likewise, in the case of *Pinus massoniana*, the vascular area of secondary needles is larger than that of primary needles [36], and secondary needles demonstrate enhanced drought resistance when subjected to drought stress treatments [37]. In the present study, the phyllodes possessed considerable vascular bundles, including central vascular bundles symmetrically distributed above and below and multiple smaller bundles in comparison to compound leaves (Figure 2b). Therefore, more vascular bundles are distributed in the phyllodes, which can effectively transport water and organic molecules and maintain the balance of water and nutrients under high light conditions. Moreover, the phyllodes possess a rigid fibrous cap attached to the vascular bundle and multiple layers of lignified parenchymatous cells that develop between the fibrous cap and the epidermis, aligning with the conclusions drawn by Dong and He [23]. In general, fibrous caps and lignified cells support and protect plant organs, enhancing their mechanical strength. In *Washingtonia robusta*, the fibrous cap’s presence significantly improved the stem’s wind resistance [38]. In fact, the fibrous cap and lignified cells are composed of several layer cells of thickened cell walls, which correlate with lignin and cellulose levels. The phyllodes exhibited higher levels of lignin and cellulose, indicating their greater rigidity in terms of physical properties. This solid mechanical organization may render them less susceptible to mechanical damage.

### 3.3. Variances in Anatomical Structure and Photosynthetic Pigment Levels Impact the Light Utilization Capacity of A. melanoxylon Heteroblastic Leaves

The light response curve reflects plants’ adaptability to light [39]. In this study (Table 3), compound leaves’ low LSP and LCP values suggested their enhanced capability to thrive in low light conditions and better adaptation to weak light environments. The reduced P′_max_ and R_d_ of compound leaves suggest a constrained photosynthetic capacity and the capability to lower respiration rate to conserve material and energy in low light environments. In addition, the WUE of compound leaves was lower than that of phyllodes, which was consistent with the results of Brodribb (1993) [18]. Therefore, it is the strong photosynthetic capacity and effective water use of phyllodes that allow them to be exposed to sunlight, while the relatively weak photosynthetic capacity and water use capacity of the compound leaves are more favorable in weak light. Similar findings were observed in *S. vulgaris*, where seedlings are frequently located beneath the canopy of a nurse plant, predominantly exhibiting needle leaves. Compared with the juvenile leaves (needle leaves), the mature leaves (scale leaves) had higher photosynthetic efficiency, more substantial tolerance to photoinhibition, and higher water use efficiency [3]. This study revealed no significant difference in Gs between phyllodes and compound leaves. However, compound leaves exhibited a higher Ci concentration and a weaker photosynthetic rate (Table 3). It is reasonable to infer that the low photosynthetic rate and unassimilated Ci in compound leaves are not caused by stomatal factors but rather may be linked to the photosynthetic enzyme activity under weak light [40]. Other studies indicated that elevated intercellular carbon dioxide concentration may regulate stomatal closure to prevent excessive transpiration and avoid cellular dehydration [41]. In compound leaves, further accumulation of CO_2_ may eventually lead to stomatal closure and restrict photosynthesis.

Moreover, the levels of chlorophyll a and carotenoids were significantly higher in compound leaves than in phyllodes in this study (Figure 5b,e). It can be deduced that compound leaves maintain growth and development by increasing chlorophyll a content to enhance photosynthetic capacity despite demonstrating greater photosynthetic activity in phyllodes than in compound leaves, as indicated by the light response curve results. The phenomenon was also observed after shading in *Carpinus Betulu* [42]. The synthesis of additional chlorophyll represents a physiological response by the plant to increase light energy absorption and to adapt to weak light conditions [43]. However, carotenoids can be used as non-enzymatic antioxidants in plants to remove reactive oxygen species (ROS) produced by oxidative stress [44]. The present study revealed a significantly higher carotenoid content in compound leaves than in phyllodes (Figure 5e). It may be attributed to the involvement of carotenoids in the physiological stress resistance processes of *A. melanoxylon*.

### 3.4. The Changes in Endogenous Hormones, the Oxidation–Reduction System, and Elemental Content Were Closely Linked throughout the Development of Weak Light-Induced Heteroblastic Leaves in A. melanoxylon

From the present results, it can be observed that the content of GA, CTK, ABA, Pro, K, Cu, Mg, and Ca, as well as the activities of SOD, POD, and CAT, are higher in the compound leaves, while the IAA and Fe content are higher in the phyllodes. According to the correlation analysis, hormone levels (GA, CTK, and ABA), antioxidant enzyme activities (SOD, POD, and CAT), and element contents (Cu, Mg, and Ca) are positively correlated with each other, but they are all negatively correlated with the photosynthetic rate (Figure 7). It is thus inferred that these substances are closely related and may collectively influence various processes of photosynthesis. Studies have shown that under stress conditions, plant hormones can increase the activities of antioxidant enzymes and osmotic regulatory substances, thereby alleviating the damage caused by ROS to cell membranes and organelles. In young seedlings of *Platycladus orientalis*, exogenous ABA, by regulating ROS metabolism, increased the activities of SOD, POD, CAT, and Pro content, significantly alleviating oxidative stress under H_2_O_2_ stress [45]. Foliar application of gibberellin (GA_3_) can increase the activities of SOD, CAT, and POD in *Abelmoschus esculentus* under salt stress, reducing the adverse effects of NaCl [46]. Supplementing CTK can enhance the drought resistance of *Agrostis stolonifera* leaves, increasing their SOD, CAT, and POD activities [47]. Finally, foliar application of CK and GA_3_ can increase the proline content in *Vigna radiata*, reducing the damage caused by waterlogging [48]. Therefore, GA, CTK, and ABA can all increase the activities of antioxidant enzymes and osmotic regulatory substances in plants. The high activity of SOD, POD, CAT, and high Pro content in the compound leaves could be attributed to the regulation of GA, CTK, and ABA. It is precisely because this regulation enhances the antioxidant mechanism and osmotic regulatory capacity of the compound leaves while clearing excess ROS and maintaining cell osmotic homeostasis that it ultimately allows for the normal functioning of photosynthesis in the compound leaves, although their photosynthetic capacity is still limited.

The content of plant hormones is closely related to the content of mineral elements. ABA can promote the opening of the guard cell outward to rectify potassium channels (GORK), leading to the efflux of potassium ions, causing changes in ion concentrations inside and outside the guard cells, and ultimately resulting in stomatal closure [49]. Ca^2+^ typically serves as a second messenger for intracellular and intercellular signal transduction and plays a crucial role in stomatal movement. For example, calcium-binding proteins (calcium-dependent protein kinases, CDPKs) can accelerate signal transduction during ABA-induced stomatal movement [50]. Iron (Fe) and Copper (Cu) both participate in the transfer and transmission of electrons, primarily binding to cytochrome proteins (plastocyanin, Fe-S clusters), driving proton pump activity, and ultimately promoting ATP synthesis [51]. Mg also plays a particularly important role in photosynthesis, as it can chelate with chlorophyll molecules, ATP, and Rubisco enzyme to form stable molecular structures [52]. However, Fe, Cu, and Mg absorption and transport require hormone involvement. Research has shown that ABA can regulate the main transcription factor (SPL7) involved in the copper deficiency response and affect the expression of its target protein (COPT), enhancing copper absorption and transport, ultimately alleviating damage caused by copper deficiency [53]. DELLAs inhibitors, which are involved in gibberellin (GA) signaling, inhibit the transcriptional activity of FIT, thereby activating the expression of iron uptake genes in the root epidermis [54]. ABA also participates in the signal response of Arabidopsis thaliana to long-term magnesium toxicity (MgT) [55]. Therefore, the accumulation of GA, CTK, ABA, K, Ca, Cu, and Mg in compound leaves may, to some extent, regulate stomatal movement and enhance light capture, energy transfer, storage, and CO_2_ assimilation capacity. The accumulation of Fe in petioles may primarily promote ATP synthesis.

In addition, endogenous hormones also play crucial roles in regulating the morphogenesis of heteroblastic leaves. Researchers demonstrated that GA prompts the development of aquatic leaves, whereas ABA prompts the growth of floating leaves in *Callitriche heterophylla* [5]. IAA and CTK play essential roles in regulating the morphological development of compound leaves in *Solanum lycopersicum* [56,57]. Previous studies have found that the shoot apical meristem of *A. melanoxylon* transforms phyllodes into compound leaves after GA treatment [58]. The present study’s findings further revealed increased levels of GA, CTK, and ABA in compound leaves, whereas phyllodes exhibited elevated IAA content (Figure 4a–d). These results imply hormones’ regulatory role in forming heteroblastic leaves. Figure 8 provides a summary and detailed comparison of different indicators between compound leaves and phyllodes, along with their potential interaction relationship.

## 4. Materials and Methods

### 4.1. Plant Materials and Weak Light Treatments

The study was conducted in an experimental field at the Research Institute of Tropical Forestry, Chinese Academy of Forestry (23° 19′ N, 113° 39′ E, 60 m elevation), Guangzhou, Guangdong Province, China. The site belongs to the marine subtropical monsoon climate, with an annual temperature of 23.2 °C and an annual total precipitation of 1891.9 mm (https://www.tianqi24.com/ (accessed on 8 June 2022)). Throughout the treatment period, the photosynthetic photon flux density (PPFD) in typical weather conditions was about 1000~1600 μmol·m^−2^·s^−1^ at 12 a.m. and 100~400 μmol·m^−2^·s^−1^ at 6 p.m., with a day/night ratio of around 11:13.

In December 2021, *A. melanoxylon* cultivated variety SR17 tissue cultured seedlings were grown in a nutrient pot (20 cm diameter, 22 cm height) with a substrate mixture of peat soil/vermiculite (1:1, *v*/*v*, pH 6.5). After one month, we chose sixty uniform and healthy seedlings (7–8 cm in height) for the weak light treatment. The weak light treatments (WL, 40% of normal light) were conducted with the seedlings wrapped in a shading net, and the control (Cont., 100% normal light) had no shading net. The seedlings from each treatment were cultured for 90 d under artificially maintained water and nutrient conditions before being harvested. Each treatment had three replicates, and each replicate consisted of ten seedlings. After being cultured for 90 d, the phyllodes in the control and the compound leaves in the weak light treatment were harvested as samples to determine the content of physiological and biochemical substances and structure observation.

### 4.2. Plant Growth Measurement

Before treatment, the height (H0), ground diameter (D0), and leaf morphology were measured. Thereafter, the height and ground diameter of the plant and the number of each type of heteroblastic leave were recorded once every thirty days. Each treatment had three replicates, and each replicate consisted of ten seedlings. The plant height was measured with a ruler from the base of the seedling to the apical bud, and the ground diameter was measured with a vernier scale. After 90 days of treatments, the plants were harvested and separated into roots, stems, compound leaves, and phyllodes, and their fresh weights were recorded. Then, their dry weights were recorded after drying at 80 °C in an oven for 48 h. Each treatment had three replicates, and each replicate consisted of three seedlings.

### 4.3. Leaf Anatomy and Stomatal Traits

Leaf anatomy was performed on the fully expanded leaves from the 3rd to 4th nodes. Small segments (about 3–5 mm long) were excised from the middle portion using a double-edged razor blade and fixed in 5% formalin, 5% acetic, and 90% alcohol (FAA). The paraffin section was made with a thickness of 8 µm, stained with toluidine blue O staining, and sealed with neutral resin. The sections were observed using a light microscope (Leica DM2000 LED, Wetzlar, Germany).

To observe the surface of the heteroblastic leaves, the middle part of the fresh leaves was intercepted, and the intercepted tissue block was not more than 3 mm^2^. The sample’s surface was gently rinsed with PBS (Servicebio, G0002, Wuhan, China) and quickly put into the electron microscope fixative (Servicebio, G1102, Wuhan, China) to be fixed at room temperature for 2 h, then transferred to 4° for storage. The samples were dehydrated sequentially with ethanol series (30%-50%-70%-80%-90%-95%-100%), dried at the critical point, fixed on SEM aluminum roots with double-sided carbon tape, and coated with gold. Images were captured with a Hitachi SU8100 scanning electron microscope at 3 kV. Stomatal density (number of stomata mm^−2^) was obtained by measuring the area of a rectangular selection free of veins and counting the number of stomata in the section using ImageJ 1.53k, then dividing the number of stomata by the area. Stomatal length, width, stomatal aperture width, and length were determined according to [59]. Each treatment had three replicates, and each replicate consisted of two replicate areas or two stomata with six readings.

### 4.4. Determination of Mineral Element Content

The dried plant samples were ground to powder and passed through a 200-mesh nylon sieve. After the sample was digested in H_2_SO_4_:H_2_O_2_, total nitrogen (N), phosphorus (P), and potassium (K) were determined using the Kjeldahl method [60], Mo-Sb colorimetric method [61], and flame photometric method [62]. The samples were digested in H_2_SO_4_:HClO_4_, and the contents of calcium (Ca), magnesium (Mg), iron (Fe), Copper (Cu), and zinc (Zn) were determined via atomic absorption spectrophotometer [63]. Boron (B) was determined via curcumin colorimetry [64].

### 4.5. Determination of Cell Wall Component Contents

After oven-drying to constant weight, 0.5 g leaves of each sample were milled into a fine powder. The hemicellulose, cellulose, and lignin contents were detected using the traditional Van Soest method [65]. The relative error of the three repeated chemical measurements of each sample was controlled to be lower than 5%.

### 4.6. Determination of Antioxidant Enzyme Activity and Pro Content

According to [66], the photochemical reduction method with nitro tetrazolium blue chloride (NBT) was used to measure SOD activity. POD activity was assessed using hydrogen peroxide (H_2_O_2_) based on guaiacol oxidation at 470 nm; the loss of H_2_O_2_ at 240 nm was used to measure CAT activity [67]. The 3% (*w*/*v*) aqueous sulfosalicylic acid Pro content was extracted and measured at 520 nm using the ninhydrin reagent [68].

### 4.7. Measurement of the Light Response Curve

The light response curves were performed on the fully expanded leaves from the 3rd to 4th nodes using the LICOR-6800 portable photosynthesizer (Li-COR, Lincoln, Dearborn, MI, USA) on consecutive sunny days after treatment for 90 d. The CO_2_ concentration in the reference chamber was set at 400 μmol·mol^−1^, the temperature was 25 °C, the RH was 60%, the gas flow rate was 500 μmol·m^−2^·s^−1^, Press Valve was 0.1 kPa, and the intensity gradient of photosynthetically active radiation (PAR) was 2000, 1800, 1600, 1400, 1200, 1000, 800, 600, 400, 200, 100, 80, 50, 20, 0 (μmol·m^−2^·s^−1^). The leaves to be tested were induced at a light intensity of 2000 μmol·m^−2^·s^−1^ for 20 min before measurement, and one point was automatically recorded every 2 min. The maximum net photosynthetic rate (P′_max_), light saturation point (LSP), light compensation point (LCP), dark respiration rate (R_d_), and dark respiration rate (R_d_) were fitted using the modified model of the rectangular hyperbola of [39]. The gas-exchange parameters, such as intercellular CO_2_ concentration (Ci), transpiration rate (Tr), stomatal conductance (Gs), and water use efficiency (WUE = Pn/Tr), are based on data measured at 2000 μmol·m^−2^·s^−1^ PAR. Each treatment had three replicates, and each replicate consisted of three seedlings.

### 4.8. Determination of Photosynthetic Pigment Contents

The photosynthetic pigments chlorophyll a, chlorophyll b, and carotenoid content were determined on the fully expanded leaves from the 3rd to 4th nodes. Approximately 0.1 g of fresh leaf samples were ground into powder with liquid nitrogen, extracted with 10 mL of 80% acetone, incubated for 24 h, and kept from light [69]. The absorbance (A) of supernatants was then measured at 470, 645, and 662 nm with a VIS scanning spectrophotometer (Helios Epsilon, Thermo Fisher Scientific, Waltham, MA, USA). Each treatment had three replicates, and each replicate consisted of five seedlings. The contents of chlorophyll a (Chl a), chlorophyll b (Chl b), total chlorophyll (Chl C), and carotenoids (Car) were determined according to the following equations:Chla (mg·mL^−1^) = 11.75 A662 − 2.35 A645(1)
Chlb (mg·mL^−1^) = 18.61 A645 − 3.96 A662(2)
ChlC (mg·mL^−1^) = Chla + Chlb(3)
Car (mg·mL^−1^) = (1000 A470 − 2.27 Chla − 81.4 Chlb)/227(4)
Pigment contents (mg·g^−1^) = (C × V)/(W × 1000)(5)

### 4.9. Determination of Endogenous Hormone Levels

The contents of endogenous hormones (IAA, CTK, GA, and ABA) in heteroblastic leaves were measured via enhanced double antibody sandwich enzyme-linked immunosorbent assay (ELISA) according to the kit’s instructions (Shanghai Enzyme-linked Biotechnology, Shanghai, China). The liquid nitrogen-ground frozen leaves (0.20 g) were dissolved in 2 mL of phosphate-buffered saline (10 mM, pH 7.5). After centrifugation at 10,000× *g* for 10 min, the supernatant (50.00 μL) was recovered. The collected supernatant (samples) and standards were applied to the microtiter plate in duplicate wells and incubated at 37 °C for 30 min. After removing the liquid, a washing buffer was used five times to wash the plates. After adding 50 µL of enzyme conjugate liquid to each well and incubating them for 30 min at 37 °C, the plates were repeatedly cleaned with washing buffer. Subsequently, 50 µL of color reagents A and B were added to the wells and incubated for 10 min at 37 °C. Ultimately, the reaction was stopped by adding 50 µL of color reagent C. The absorbance (OD value) of samples and standards was measured at 450 nm, and the standard curve was calculated according to the concentration and OD value of the standards. Finally, the standard curve’s regression equation was used to calculate the concentration of each hormone. Each treatment had three replicates, and each replicate consisted of three seedlings.

### 4.10. Statistical Analysis

The results are reported as averages with standard errors (SD). Microsoft Excel (2016) was used to perform the preliminary statistics. Then, data were analyzed using one-way ANOVA, and the means were separated with Duncan’s multiple range test at the 5% probability level using SPSS statistical package version 22.0 (IBM Corp., Amonk, NY, USA). The chart class was produced with Origin 8.5 (OriginLab Corporation, Northampton, MA, USA).

## 5. Conclusions

This study provided new information for developing heteroblastic leaves of *A. melanoxylon* seedlings induced by weak light. Weak light limits the growth of the seedlings, but leaves remain juvenile in shape (compound leaves), while the seedlings under the control treatment grow mature leaves (phyllodes). This difference also means that compound leaves and phyllodes have different adaptability. Secondly, these two kinds of leaves are harmonized in their morphological characteristics and physiological characteristics so that phyllodes can adapt to the sun and compound leaves can resist the adverse effects of the weak light environment. The above research results also provide a basis for understanding the diverse survival strategies that heteroblastic plants employ to adapt to environmental changes.

## Figures and Tables

**Figure 1 plants-13-00870-f001:**
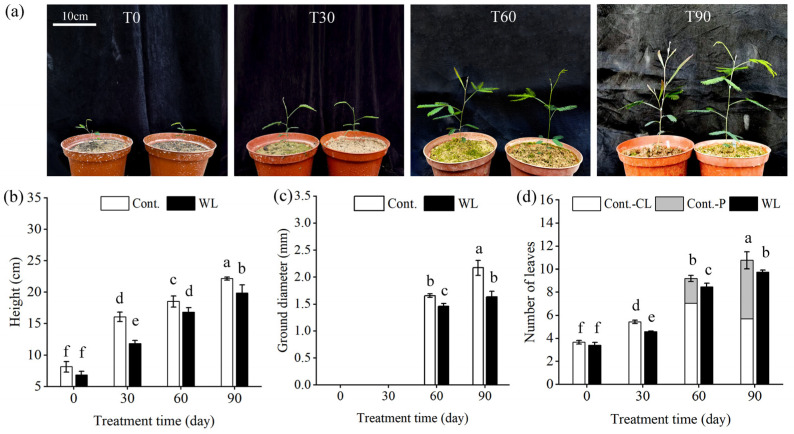
The phenotype and growth characteristics of *A. melanoxylon* seedlings under weak light treatment. (**a**) The phenotype of *A. melanoxylon* seedlings throughout weak light treatment, the seedlings on the left of the photo are the seedlings in the control, and the right one shows the seedlings under weak light treatment; (**b**) Height, (**c**) ground diameter, and (**d**) number of leaves of the seedlings under weak light and the control treatment, WL, the weak light treatment; CL, the compound leaves; P, the phyllodes; T0, T30, T60, and T90 mean the seedlings treated for 0 d, 30 d, 60 d, and 90 d, respectively; different lowercase letters in each column indicate significant differences between treatments (*p* < 0.05).

**Figure 2 plants-13-00870-f002:**
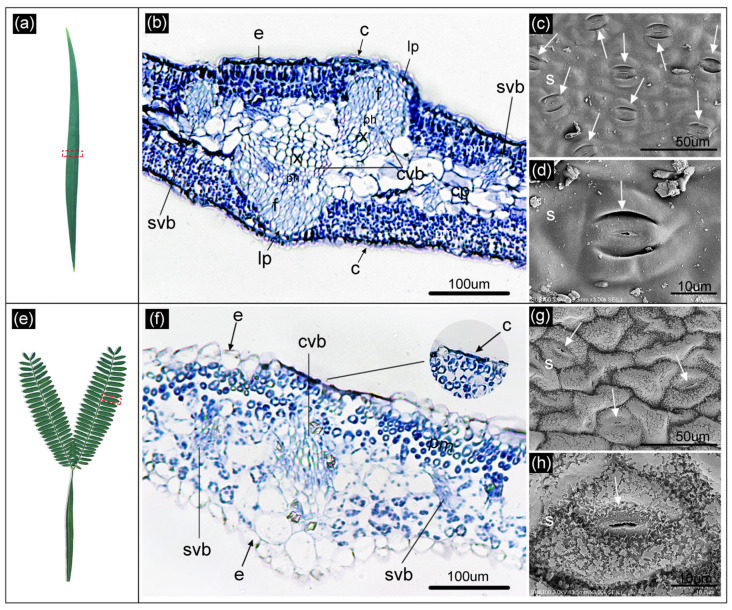
The anatomical structure of *A. melanoxylon* heteroblastic leaves. (**a**), Phyllode (red region represents sample range); (**b**), phyllode transverse sections; (**c**), phyllode adaxial surface; (**d**), phyllode stomata; (**e**), compound leaf (red region represents sample range); (**f**), compound leaf transverse sections; (**g**), compound leaf adaxial surface; (**h**), compound leaf stomata. Abbreviations: c, cuticle; cp, central parenchyma; cvb, central vascular bundle; e, epidermal cell; f, fiber; lp, lignified parenchyma; mvb, marginal vascular bundle; ph, phloem; pm, palisade mesophyll; s, stomata; svb, small vascular bundle; x, xylem.

**Figure 3 plants-13-00870-f003:**
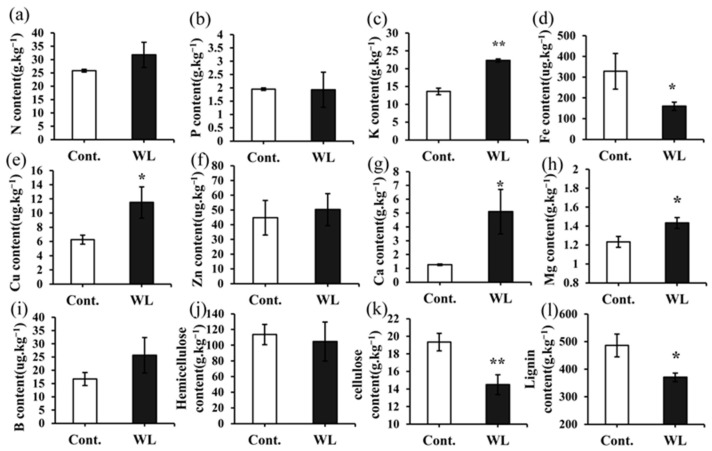
The content of nutrient elements, lignin, cellulose, and hemicellulose in *A. melanoxylon* heteroblastic leaves. (**a**) Nitrogen, (**b**) phosphorus, (**c**) potassium, (**d**) iron, (**e**) cuprum, (**f**) zinc, (**g**) calcium, (**h**) magnesium, (**i**) boron, (**j**) hemicellulose, (**k**) cellulose, and (**l**) lignin contents in *A. melanoxylon* heteroblastic leaves under weak light and the control treatment. *, significant differences at *p*-value < 0.05 (*n* = 3), **, significant differences at *p*-value < 0.01 (*n* = 3).

**Figure 4 plants-13-00870-f004:**
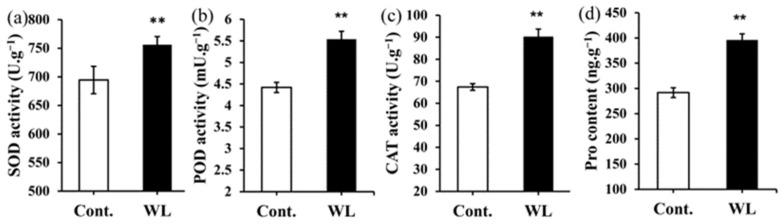
Differences in antioxidant enzyme activities and the content of osmotic regulatory substances in *A. melanoxylon* heteroblastic leaves. (**a**) SOD, (**b**) POD, and (**c**) CAT activities, and (**d**) Pro content in *A. melanoxylon* heteroblastic leaves under weak light and the control treatment. **, significant differences at *p*-value < 0.01 (*n* = 5).

**Figure 5 plants-13-00870-f005:**
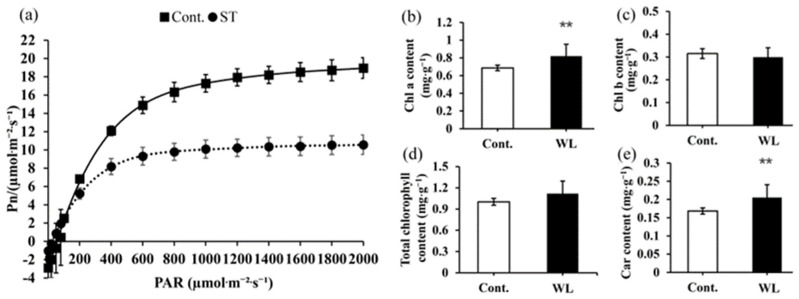
Light response curve and photosynthetic pigment content of *A. melanoxylon* heteroblastic leaves. (**a**) Light response curve and (**b**) chlorophyll a, (**c**) chlorophyll b, (**d**) total chlorophyll, and (**e**) carotenoid content in *A. melanoxylon* heteroblastic leaves under weak light and the control treatment. **, significant differences at *p*-value < 0.01 (*n* = 6).

**Figure 6 plants-13-00870-f006:**
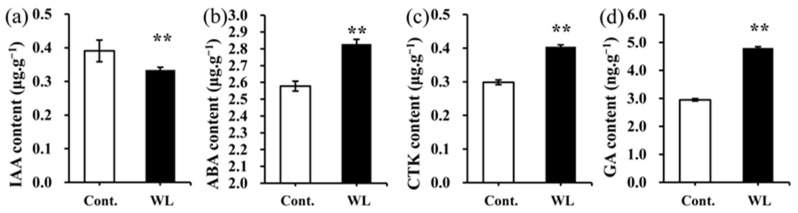
Endogenous hormone levels in *A. melanoxylon* heteroblastic leaves. (**a**) IAA, (**b**) ABA, (**c**) CTK, and (**d**) GA content in *A. melanoxylon* heteroblastic leaves under weak light and the control treatment. **, significant differences at *p*-value < 0.01 (*n* = 5).

**Figure 7 plants-13-00870-f007:**
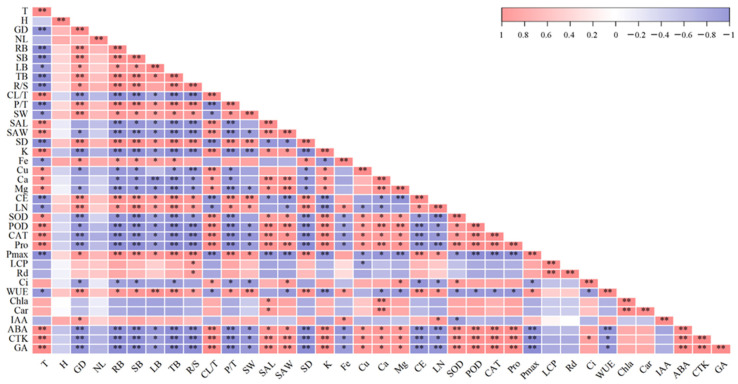
Correlation analysis among the comparison indexes of *A. melanoxylon* heteroblastic leaves. *, significant correlation at *p*-value < 0.05; **, significant correlation at *p*-value < 0.01. T, treatment; H, height; GD, ground diameter; NL, number of leaves; RB, root biomass; SB, stem biomass; LB, leaf biomass; TB, total biomass; SW, stomatal width; SAL, stomatal aperture length; SAW, stomatal aperture width; SD, stomatal density; CE, cellulose content; LN, lignin content.

**Figure 8 plants-13-00870-f008:**
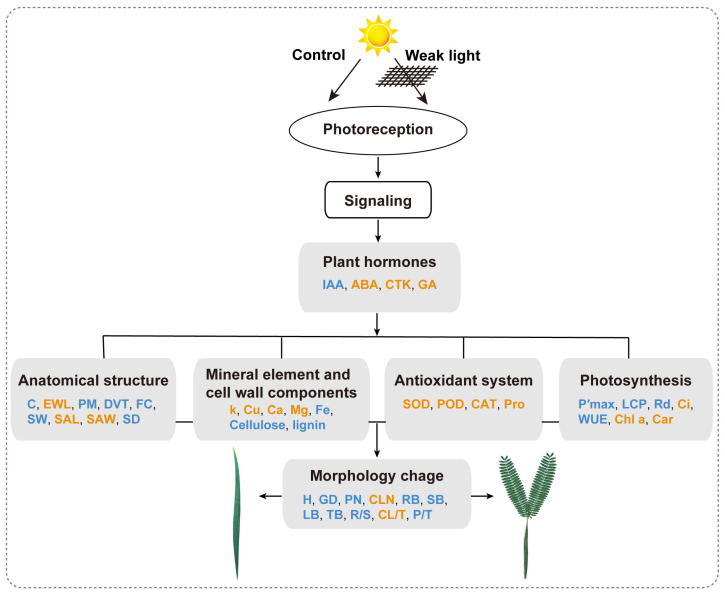
Differences in the development and physiology of compound leaves and phyllodes in *A. melanoxylon*. The orange fonts represent significantly or extremely significantly higher values that were in compound leaves, while the blue fonts represent significantly or extremely significantly lower values in compound leaves. H, plant height; GD, ground diameter; PN, phyllodes number; CLN, compound leaves number; RB, root biomass; SB, stem biomass; LB, leaf biomass; TB, total biomass; R/S, the ratio of root to stem; CL/T, the ratio of compound leaves to total biomass; P/T, the ratio of phyllodes to total biomass; C, cuticle; EWL, epidermal waxy layer; PM, palisade mesophyll; DVT, developed vascular tissue; FCs, fiber caps; SW, stomatal width; SAL, stomatal aperture length; SAW, stomatal aperture width; SD, stomatal density; P′_max_, the maximum net photosynthetic rate; LCP, light compensation point; R_d_, dark respiration rate; Ci, intercellular CO_2_ concentration; WUE, water use efficiency.

**Table 1 plants-13-00870-t001:** The differences in biomass distribution between seedlings under weak light treatment in the control.

Treatment	Root Biomass (g)	Stem Biomass (g)	Leaf Biomass (g)	Total Biomass (g)	R/S	CL/T	P/T
Cont.	0.94 ± 0.05 a	0.35 ± 0.04 a	1.04 ± 0.18 a	2.34 ± 0.15 a	0.67 ± 0.08 a	0.24 ± 0.05 b	0.21 ± 0.02 a
WL	0.16 ± 0.005 b	0.11 ± 0.03 b	0.76 ± 0.14 b	1.02 ± 0.14 b	0.19 ± 0.04 b	0.74 ± 0.04 a	0.00 ± 0.00 b

R/S, the ratio of root to stem biomass; CL/T, the ratio of compound leaves to total biomass; P/T, the ratio of phyllodes to total biomass. The data in the table are the means ± SD (*n* = 3), different lowercase letters after the numbers indicate significant differences between treatments (*p* < 0.05).

**Table 2 plants-13-00870-t002:** The stomatal traits of *A. melanoxylon* heteroblastic leaves.

Leaf Type	Stomatal Length (μm)	Stomatal Width (μm)	Stomatal Aperture Length (μm)	Stomatal Aperture Width (μm)	Stomatal Density (number·mm^−2^)
Cont.	14.75 ± 1.10 a	9.07 ± 0.53 a	2.78 ± 1.09 b	0.91 ± 0.36 b	660 ± 70.25 a
WL	15.46 ± 2.54 a	6.87 ± 1.16 b	6.0 ± 1.03 a	1.41 ± 0.27 a	272 ± 35.12 b

The data in the table are the means ± SD (*n* = 6); different lowercase letters after the numbers indicate significant differences between the two types of leaves (*p* < 0.05).

**Table 3 plants-13-00870-t003:** Photosynthetic parameters of *A. melanoxylon* heteroblastic leaves.

Leaf Type	P′_max_ (μmol·m^−2^·s^−1^)	LSP (μmol·m^−2^·s^−1^)	LCP (μmol·m^−2^·s^−1^)	R_d_ (μmol·m^−2^·s^−1^)
Cont.	18.76 ± 0.79 a	1748.21 ± 241.59 a	51.48 ± 27.3 a	3.54 ± 2.63 a
WL	10.56 ± 1.14 b	1509.91 ± 144.77 a	25.68 ± 6.04 b	1.28 ± 0.32 b
**Leaf Type**	**Ci** **(µmol·mol^−1^)**	**Tr** **(mol·m^−2^·s^−1^)**	**Gs** **(mol·m^−2^·s^−1^)**	**WUE** **(μmol·mol^−1^)**
Cont.	174.95 ± 46.66 b	0.0042 ± 0.0006 a	0.159 ± 0.03 a	4488.87 ± 519.35 a
WL	245.86 ± 15.89 a	0.0038 ± 0.0005 a	0.138 ± 0.02 a	2762.18 ± 502.68 b

P′_max_, the maximum net photosynthetic rate; LSP, light saturation point; LCP, light compensation point; R_d_, dark respiration rate; Ci, intercellular CO_2_ concentration; Tr, transpiration rate; Gs, stomatal conductance; WUE, water use efficiency. Different lowercase letters after the numbers indicate significant differences between the two types of leaves (*p* < 0.05) (*n* = 6).

## Data Availability

The original contributions presented in the study are included in the article material. Further inquiries can be directed to the corresponding author.

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
