# Peer review of "Morphological, Anatomical, and Physiological Characteristics of Heteroblastic Acacia melanoxylon Grown under Weak Light"

_plants, 2024, doi:10.3390/plants13060870_

Round 1

Reviewer 1 Report

Comments and Suggestions for Authors

This paper reports on the characteristics of two types of leaf morphology observed in Acacia melanoxylon, commonly known as Australian blackwood, under different natural light intensities. By comparing plant morphology, leaf anatomy, metal contents, antioxidant activities, photosynthetic activity, and photosynthetic pigment contents, the authors describe the characteristics of heteroblastic leaves observed under shaded light growth conditions. I believe the results are worthy of publication. However, I suggest the authors address the following points before publication:

In the manuscript, the authors propose a model (Fig. 8) to account for the heteromorphic development observed in A. melanoxylon. A major weakness of the current paper is that this model and the conclusions drawn are not sufficiently justified by the results presented. Particularly, there is a lack of exploration into the effects of light, despite acknowledging its significant impact on heteromorphic development. The study demonstrates only a reduction in natural sunlight and fails to confirm which aspect of light, such as shading threshold or required light wavelengths, controls leaf development. Typically, blue and infrared lights are crucial for plant development, and their reduction might cause the observed differences. The discrepancy in the growth field is also problematic, as the natural habitat of these plants is in Australia, where the intensity and light spectrum differ from China, an aspect which lacks ecological consideration. If this study is viewed as experimental botanical research, the use of the term "shading" of natural sunlight renders the study somewhat premature.

I suggest reconsidering the scheme presented in Fig. 8. Since light clearly impacts leaf morphology in this study, light perception mechanisms, including chlorophylls, phytochromes, and photoreceptors, should be placed at the top of the cascade. This could be followed by signal transduction processes, metabolic changes including plant hormones, structural changes involving the binding of divalent cations, and enzymatic upregulations. As discussed in the manuscript, ROS formed during exposure to full sunlight may be key to inducing developmental or morphological changes. The ROS H2O2, which can diffuse across membranes, is considered a universal signaling molecule regulating gene expressions in response to abiotic and biotic stresses, not only in plants but also in animals and bacteria. It is highly plausible that photoinhibiting conditions could result in H2O2 formation in cells, leading to leaf developmental changes in A. melanoxylon. In this context, exploring photoinhibition at high light intensity in both types of leaves is necessary. A partial shading treatment for the leaves could also be interesting to investigate the mechanism.

Despite the potential issues mentioned, the results presented in this paper are insightful and will contribute to future research in relevant fields. I recommend the authors restructure the manuscript to emphasize the characterization of the heteroblastic leaves more. Additionally, it would be better to refer to the model mentioned in the abstract (Fig. 8) as a hypothetical scheme.

<Title>

Simplifying the title to something like "Morphological, Anatomical, and Physiological Characteristics of Heteroblastic Acacia melanoxylon Grown Under Weak Light" would be better.

<Abstract>

Please include the conclusions of this study. Phrases like "a model was proposed" or "this study offers insight" are not sufficient as conclusions.

<Materials and Methods>

The environmental data is missing. Please include conditions of natural light such as light intensity and day/night ratio.

Author Response

Dear reviewer,

Thank you for giving us the opportunity to submit a revised draft of the manuscript (ID: 2884787) for publication in Plants. We appreciate your time and effort in providing valuable feedback on our manuscript. We have revised the manuscript extensively based on the insightful comments and valuable improvements you provided. The changes have been highlighted in red color in the revised manuscript. Please find the point-by-point response to your comments and concerns below.

Q1: In the manuscript, the authors propose a model (Fig. 8) to account for the heteromorphic development observed in A. melanoxylon. A major weakness of the current paper is that this model and the conclusions drawn are not sufficiently justified by the results presented. Particularly, there is a lack of exploration into the effects of light, despite acknowledging its significant impact on heteromorphic development. The study demonstrates only a reduction in natural sunlight and fails to confirm which aspect of light, such as shading threshold or required light wavelengths, controls leaf development. Typically, blue and infrared lights are crucial for plant development, and their reduction might cause the observed differences. The discrepancy in the growth field is also problematic, as the natural habitat of these plants is in Australia, where the intensity and light spectrum differ from China, an aspect which lacks ecological consideration. If this study is viewed as experimental botanical research, the use of the term "shading" of natural sunlight renders the study somewhat premature.

A1: Thank you for your professional advice. Figure 8 showed the differences in various indicators between the two kinds of leaves revealed in detail in this study and the potential interaction between these indicators according to existing theoretical knowledge. This article initially intended to focus on the differences in the characteristics of the heteroblastic leaves, and we will investigate the formation mechanism of the heteroblastic leaves in the future. It is true that because the parameters of the shading threshold or required light wavelengths are not clear, the artificial light control test cannot be performed well. So, we used the shading net to process the weak light conditions in the field. The light under the net during the treatment was about 40% of that of the control, and there was no drastic change in light quality, such as the ratio of red to blue light. According to our measurements, the red-blue ratio was approximately 1.3 under the shading net and 1.2 in the field (control), whereas it reached 1.8 under tree shade during the experiment. It was determined that the light quality was affected differently by the shading net we employed in contrast to the shade provided by trees. In brief, PPFD values were used as treatment and control parameters in this study. Based on this study, we will further detail manual control experiments and investigate the effects of light quality on heteroblastic leaf development. The term "shade" was deemed unsuitable, and consequently, we revised it to "weak light."

Q2: I suggest reconsidering the scheme presented in Fig. 8. Since light clearly impacts leaf morphology in this study, light perception mechanisms, including chlorophylls, phytochromes, and photoreceptors, should be placed at the top of the cascade. This could be followed by signal transduction processes, metabolic changes including plant hormones, structural changes involving the binding of divalent cations, and enzymatic upregulations. As discussed in the manuscript, ROS formed during exposure to full sunlight may be key to inducing developmental or morphological changes. The ROS H2O2, which can diffuse across membranes, is considered a universal signaling molecule regulating gene expressions in response to abiotic and biotic stresses, not only in plants but also in animals and bacteria. It is highly plausible that photoinhibiting conditions could result in H2O2 formation in cells, leading to leaf developmental changes in A. melanoxylon. In this context, exploring photoinhibition at high light intensity in both types of leaves is necessary. A partial shading treatment for the leaves could also be interesting to investigate the mechanism.

A2: Thank you for your professional advice. In this study, we did not detail the results of light perception mechanisms and the signal transduction processes. In reference to your suggestions, Figure 8 has been appropriately modified to demonstrate the comparative analysis of various indicators of the two types of leaves rather than a model to form the mechanism. Thank you for your suggestion of H2O2 signaling and partial shading treatment. It provides a very professional idea for the analysis of the formation mechanism.

Q3: Despite the potential issues mentioned, the results presented in this paper are insightful and will contribute to future research in relevant fields. I recommend the authors restructure the manuscript to emphasize the characterization of the heteroblastic leaves more. Additionally, it would be better to refer to the model mentioned in the abstract (Fig. 8) as a hypothetical scheme.

A3: Thank you for your recognition. The abstract has been revised (lines 27-29). The statements about the formation mechanism discussed in the results have been removed, and the discussion has been modified to emphasize the differences and correlations between various characteristics of heteroblastic leaves.

Q4: <Title> Simplifying the title to something like "Morphological, Anatomical, and Physiological Characteristics of Heteroblastic Acacia melanoxylon Grown Under Weak Light" would be better.

A4: Thanks for your suggestion. It has been modified.

Q5: <Abstract> Please include the conclusions of this study. Phrases like "a model was proposed" or "this study offers insight" are not sufficient as conclusions.

A5: This sentence has been deleted: "Based on these results, a model was proposed to explain the observed differences. This study offers insights into the mechanism of heteroblastic leaf formation and establishes a basis for understanding the diverse survival strategies that plants employ to adapt to environmental changes." It was modified to "The comparative analysis of compound leaves and phyllodes provides a basis for understanding the diverse survival strategies that heteroblastic plants employ to adapt to environmental changes." (lines 27-29).

Q6: <Materials and Methods> The environmental data is missing. Please include conditions of natural light such as light intensity and day/night ratio.

A6: The light intensity and day/night ratio were added in materials and methods, lines 449-452.

Reviewer 2 Report

Comments and Suggestions for Authors

The paper entitled Insights into the differences in leaf morphological, anatomical, and physiological characteristics of heteroblastic Acacia melanoxylon induced by shading and written by Bai et al. shows the effect of shading on Acacia melanoxylon.

It is a well-written paper, with a large number of experiments that cover the objectives proposed by the researchers.  The length of the paper is correct, with an extensive and detailed discussion, and it relies on the experimental results to achieve its final objective.

However, there are a couple of points that need to be corrected or clarified.

First, in the results, it is not clear to me the number of replicates the authors are using. If you look at figure 1, the error bars should represent the deviation of the data. If we take the deviation and transform it into standard error for three samples, the data would show that there might not be significant differences.

Therefore, my question is whether the authors have used the pseudoreplicates as replicates for the calculation of the mean of the data; in short, how much data have they used for the calculation of the mean?

Secondly, in section 2.2, the authors compare the anatomical structure between the two types of leaf structures; why do they compare the stomatal density of both structures on different surfaces? Have they checked that the number of stomata is the same on the adaxial and abaxial surfaces? I do not understand why they compare the abaxial surface with the adaxial surface, because if the leaf is not anphystomatic, they could not be compared. These would be wrong results that cannot be used.

Short comments:

Page 2, line 51: write the scientific name in italics

Page 2, line 63: add a space after the citation [1].

Page 3: where Figures 1b and 1c are indicated, figure 1d should be added.

Figure 1b: why there are two letters after 60 days

Adjust table 1

Figures 2c and 2d are not mentioned in the text. If they are not necessary they should be deleted.

Figure 2: the letters in the pictures are written in lower case, and in the figure caption in upper case. Unify criteria.

In figure 2 the orientation of the cuts is different for both types of leaf structures. The same orientation should have been used in order to compare the samples more accurately.

Adjust table 2.

In all the results sections, the last paragraph is part of the discussion. It should be deleted and changed to the correct paragraph.

In Table 3, the LSP of the shaded plants is very high. If we look at figure 5a, the value would be around 600-800, and from that PAR radiation, photosynthesis is saturated.

Page 10: write the genus of p. euphratica with a capital letter.

Page 14: change Whereafter to Thereafter?

Page 15: 500 µmol-s-1 in superscript and by surface, i.e. 500 µmol-m-2-s-1

References: check all references and write the scientific names in italics.

Author Response

Dear reviewer,

Thank you for giving us the opportunity to submit a revised draft of the manuscript (ID: 2884787) for publication in Plants. We appreciate your time and effort in providing valuable feedback on our manuscript. We have revised the manuscript extensively based on the insightful comments and valuable improvements you provided. The changes have been highlighted in red color in the revised manuscript. Please find the point-by-point response to your comments and concerns below.

Q1: First, in the results, it is not clear to me the number of replicates the authors are using. If you look at figure 1, the error bars should represent the deviation of the data. If we take the deviation and transform it into standard error for three samples, the data would show that there might not be significant differences. Therefore, my question is whether the authors have used the pseudoreplicates as replicates for the calculation of the mean of the data; in short, how much data have they used for the calculation of the mean?

A1: Thank you for your professional advice. The plant height, ground diameter, and leaf morphology were assessed based on data from 3 biological replicates containing 10 seedlings. Detailed descriptions have been added to the materials and methods (lines 467-468). The results in Figure 1 were an analysis of the values of 30 samples.

Q2: Secondly, in section 2.2, the authors compare the anatomical structure between the two types of leaf structures; why do they compare the stomatal density of both structures on different surfaces? Have they checked that the number of stomata is the same on the adaxial and abaxial surfaces? I do not understand why they compare the abaxial surface with the adaxial surface, because if the leaf is not anphystomatic, they could not be compared. These would be wrong results that cannot be used.

A2: Thank you for your professional advice. The primary purpose of this study was to compare the stomatal characteristics in the adaxial surface of the two kinds of leaves. Due to the writing error of "abaxial", it has been corrected to "adaxial".

Q3: Page 2, line 51: write the scientific name in italics

A3: It has been modified as suggested.

Q4: Page 2, line 63: add a space after the citation [1].

A4: It has been modified as suggested.

Q5: Page 3: where Figures 1b and 1c are indicated, figure 1d should be added. Figure 1b: why there are two letters after 60 days

A5: Figure 1d has been added in line 111. In section 4.10, We mentioned, “the data were analyzed using one-way ANOVA, and the means were separated with Duncan’s multiple range test at the 5% probability level”. The disparity in values associated with identical letters does not hold statistical significance. There was no statistical significance observed in the values between the Control group at 60 d and the WL group at 90 d, the Control group at 60 d and the WL group at 60 d, and the WL group at 60 d and the Control group at 30 d.

Q6: Adjust table 1

A6: Thanks for your suggestion. It has been modified.

Q7: Figures 2c and 2d are not mentioned in the text. If they are not necessary they should be deleted.

A7: We added Figures 2c and 2d in section 2.2 (lines 138 and 154). The results presented in Table 2 were derived from the data in Figures 2c and 2d, further enhancing the clarity by integrating numerical data with phenotypic images.

Q8: Figure 2: the letters in the pictures are written in lower case, and in the figure caption in upper case. Unify criteria.

A8: It has been modified as suggested.

Q9: In figure 2 the orientation of the cuts is different for both types of leaf structures. The same orientation should have been used in order to compare the samples more accurately.

A9: Figures 2b and 2f were the pictures of both leaf transverse plane paraxial face types.

Q10: Adjust table 2.

A10: Thanks for your suggestion. It has been modified.

Q11: In all the results sections, the last paragraph is part of the discussion. It should be deleted and changed to the correct paragraph.

A11: It has been modified as suggested.

Q12: In Table 3, the LSP of the shaded plants is very high. If we look at figure 5a, the value would be around 600-800, and from that PAR radiation, photosynthesis is saturated.

A12: The relevant data for the light response curve in this study were calculated through the mechanical modeling of leaf photosynthesis response to light by YZP [39]. As shown in Table 3, the photosynthetic rate of shade leaves continues to increase in PAR radiation beyond 800, which was not apparent in Figure 5a because it was relatively gentle.

Q13: Page 10: write the genus of p. euphratica with a capital letter.

A13: It has been modified as suggested.

Q14: Page 14: change Whereafter to Thereafter?

A14: It has been modified as suggested.

Q15: Page 15: 500 µmol-s-1 in superscript and by surface, i.e. 500 µmol-m-2-s-1

A15: It has been modified as suggested.

Q16: References: check all references and write the scientific names in italics.

A16: It has been modified as suggested.

Round 2

Reviewer 1 Report

Comments and Suggestions for Authors

This manuscript represents a revised version of the paper previously submitted to the journal Plants. The authors have substantially improved the manuscript by incorporating all the suggested revisions and addressing the comments provided. Their responses to these comments and suggestions appear appropriate and logical, contributing to the overall enhancement of the paper.

Author Response

Thank you for your approval of our revised manuscript. Since you mentioned the significant letters in Figure 1b last time, we have re-analyzed the data in Figure 1 based on the number of repeats (n=3) this time. As a result, the SD value in Figure 1 and the significant letters of 60 d in Figure 1b have slightly changed.

Reviewer 2 Report

Comments and Suggestions for Authors

My only doubt is the number of replicates. If the analysis of the samples in figure 1 corresponds to 30 samples, will the mean and variance of the data be calculated on the basis of 30 samples?
If so, it should be corrected, as the number of replicates is 3, not 30. The 10 samples of seedlings per treatment would be pseureplicates.
Therefore you should re-analyse the data with n=3.

Author Response

Thank you for your comments. We confirmed that the data shown in Figure 1 was obtained using a sample size of n=30. So, they were re-analyzed with a repetition of 3, resulting in slight alterations to the SD value in Figure 1 and the significant letters of 60 d in Figure 1b.